# Contrastive Representation Distillation

**Yonglong Tian**
MIT CSAIL
yonglong@mit.edu

**Dilip Krishnan**
Google Research
dilipkay@google.com

**Phillip Isola**
MIT CSAIL
phillipi@mit.edu

## Abstract

Often we wish to transfer representational knowledge from one neural network to another. Examples include distilling a large network into a smaller one, transferring knowledge from one sensory modality to a second, or ensembling a collection of models into a single estimator. Knowledge distillation, the standard approach to these problems, minimizes the KL divergence between the probabilistic outputs of a teacher and student network. We demonstrate that this objective ignores important *structural* knowledge of the teacher network. This motivates an alternative objective by which we train a student to capture significantly more *information* in the teacher's representation of the data. We formulate this objective as contrastive learning. Experiments demonstrate that our resulting new objective outperforms knowledge distillation and other cutting-edge distillers on a variety of knowledge transfer tasks, including single model compression, ensemble distillation, and cross-modal transfer. Our method sets a new state-of-the-art in many transfer tasks, and sometimes even outperforms the *teacher* network when combined with knowledge distillation.

## 1 Introduction

Knowledge distillation (KD) transfers knowledge from one deep learning model (the teacher) to another (the student). The objective originally proposed by Hinton et al. (2015) minimizes the KL divergence between the teacher and student outputs. This formulation makes intuitive sense when the output is a distribution, e.g., a probability mass function over classes. However, often we instead wish to transfer knowledge about a *representation*. For example, in the problem of "cross-modal distillation", we may wish to transfer the representation of an image processing network to a sound (Aytar et al., 2016) or to depth (Gupta et al., 2016) processing network, such that deep features for an image and the associated sound or depth features are highly correlated. In such cases, the KL divergence is undefined.

Representational knowledge is *structured* – the dimensions exhibit complex interdependencies. The original KD objective introduced in (Hinton et al., 2015) treats all dimensions as independent, conditioned on the input. Let $\mathbf{y}^T$ be the output of the teacher and $\mathbf{y}^S$ be the output of the student. Then the original KD objective function, $\psi$, has the fully factored form: $\psi(\mathbf{y}^S, \mathbf{y}^T) = \sum_i \phi_i(\mathbf{y}_i^S, \mathbf{y}_i^T)^*$. Such a factored objective is insufficient for transferring structural knowledge, i.e. dependencies between output dimensions $i$ and $j$. This is similar to the situation in image generation where an $L_2$ objective produces blurry results, due to independence assumptions between output dimensions.

To overcome this problem, we would like an objective that captures correlations and higher order output dependencies. To achieve this, in this paper we leverage the family of *contrastive* objectives (Gutmann & Hyvärinen, 2010; Oord et al., 2018; Arora et al., 2019; Hjelm et al., 2018). These objective functions have been used successfully in recent years for density estimation and representation learning, especially in self-supervised settings. Here we adapt them to the task of knowledge distillation from one deep network to another. We show that it is important to work in representation space, similar to recent works such as Zagoruyko & Komodakis (2016a); Romero et al. (2014). However, note that the loss functions used in those works do not explicitly try to capture correlations or higher-order dependencies in representational space.

---

Code: http://github.com/HobbitLong/RepDistiller.
*In particular, in (Hinton et al., 2015), $\phi_i(a, b) = -a \log b \quad \forall i$

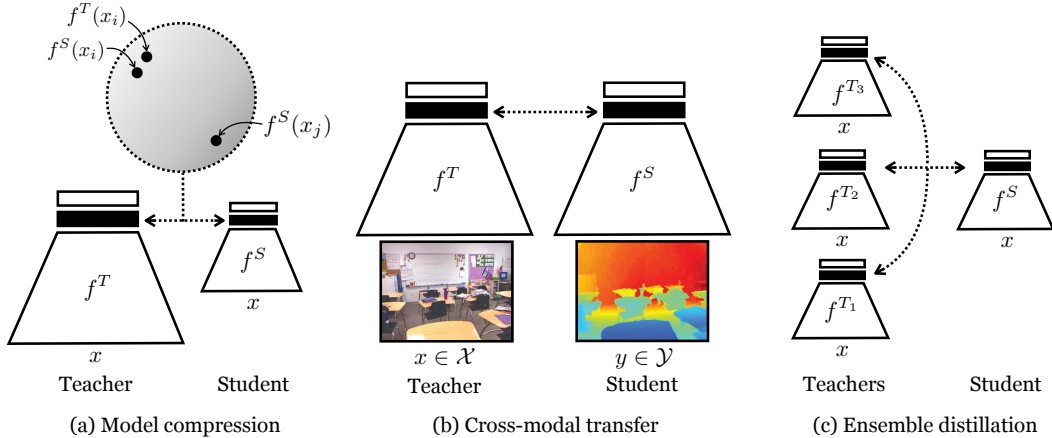

Figure 1: The three distillation settings we consider: (a) compressing a model, (b) transferring knowledge from one modality (e.g., RGB) to another (e.g., depth), (c) distilling an ensemble of nets into a single network. The constrastive objective encourages the teacher and student to map the same input to close representations (in some metric space), and different inputs to distant representations, as indicated in the shaded circle.

Our objective maximizes a lower-bound to the mutual information between the teacher and student representations. We find that this results in better performance on several knowledge transfer tasks. We conjecture that this is because the contrastive objective better transfers all the information in the teacher's representation, rather than only transferring knowledge about conditionally independent output class probabilities. Somewhat surprisingly, the contrastive objective even improves results on the originally proposed task of distilling knowledge about class probabilities, for example, compressing a large CIFAR100 network into a smaller one that performs almost as well. We believe this is because the correlations between different class probabilities contains useful information that regularizes the learning problem. Our paper forges a connection between two literatures that have evolved mostly independently: *knowledge distillation* and *representation learning*. This connection allows us to leverage strong methods from representation learning to significantly improve the SOTA on knowledge distillation. Our contributions are:

1. A contrastive-based objective for transferring knowledge between deep networks.
2. Applications to model compression, cross-modal transfer, and ensemble distillation.
3. Benchmarking 12 recent distillation methods; CRD outperforms all other methods, e.g., 57% average relative improvement over the original KD (Hinton et al., 2015) [†], which, surprisingly, performs the second best.

## 2 RELATED WORK

The seminal work of Buciluǎ et al. (2006) and Hinton et al. (2015) introduced the idea of knowledge distillation between large, cumbersome models into smaller, faster models without losing too much generalization power. The general motivation was that at training time, the availability of computation allows "slop" in model size, and potentially faster learning. But computation and memory constraints at inference time necessitate the use of smaller models. Buciluǎ et al. (2006) achieve this by matching output logits; Hinton et al. (2015) introduced the idea of temperature in the softmax outputs to better represent smaller probabilities in the output of a single sample. These smaller probabilities provide useful information about the learned representation of the teacher model; some tradeoff between large temperatures (which increase entropy) or small temperatures tend to provide the highest transfer of knowledge between student and teacher. The method in (Li et al., 2014) was also closely related to (Hinton et al., 2015).

Attention transfer (Zagoruyko & Komodakis, 2016a) focuses on the features maps of the network as opposed to the output logits. Here the idea is to elicit similar response patterns in the teacher and student feature maps (called "attention"). However, only feature maps with the same spatial

---

[†]Average relative improvement = $\frac{1}{N}\sum_{i=1}^{N}\frac{Acc_{crd}^{i}-Acc_{kd}^{i}}{Acc_{kd}^{i}-Acc_{van}^{i}}$, where $Acc_{crd}^{i}$, $Acc_{kd}^{i}$, and $Acc_{van}^{i}$ represent the accuracies of CRD, KD, and vanilla training of the $i$-th student model, respectively.

resolution can be combined in this approach, which is a significant limitation since it requires student and teacher networks with very similar architectures. This technique achieves state of the art results for distillation (as measured by the generalization of the student network). FitNets (Romero et al., 2014) also deal with intermediate representations by using regressions to guide the feature activations of the student network. Since Zagoruyko & Komodakis (2016a) do a weighted form of this regression, they tend to perform better. Other papers (Yim et al., 2017; Huang & Wang, 2017; Kim et al., 2018; Yim et al., 2017; Huang & Wang, 2017; Ahn et al., 2019; Koratana et al., 2019) have enforced various criteria based on representations. The contrastive objective we use in this paper is the same as that used in CMC (Tian et al., 2019). But we derive from a different perspective and give a rigorous proof that our objective is a lower bound on mutual information. Our objective is also related to the InfoNCE and NCE objectives introduced in (Oord et al., 2018; Gutmann & Hyvärinen, 2010). Oord et al. (2018) use contrastive learning in the context of self-supervised representations learning. They show that their objective maximizes a lower bound on mutual information. A very related approach is used in (Hjelm et al., 2018). InfoNCE and NCE are closely related but distinct from adversarial learning (Goodfellow et al., 2014). In (Goodfellow, 2014), it is shown that the NCE objective of Gutmann & Hyvärinen (2010) can lead to maximum likelihood learning, but not the adversarial objective.

## 3 METHOD

The key idea of contrastive learning is very general: learn a representation that is close in some metric space for "positive" pairs and push apart the representation between "negative" pairs. Fig. 1 gives a visual explanation for how we structure contrastive learning for the three tasks we consider: model compression, cross-modal transfer and ensemble distillation.

### 3.1 CONTRASTIVE LOSS

Given two deep neural networks, a teacher $f^T$ and a student $f^S$. Let $x$ be the network input; we denote representations at the penultimate layer (before logits) as $f^T(x)$ and $f^S(x)$ respectively. Let $x_i$ represent a training sample, and $x_j$ another randomly chosen sample. We would like to push closer the representations $f^S(x_i)$ and $f^T(x_i)$ while pushing apart $f^S(x_i)$ and $f^T(x_j)$. For ease of notation, we define random variables $S$ and $T$ for the student and teacher's representations of the data respectively:

$$x \sim p_{\text{data}}(x) \qquad \triangleleft \quad \textbf{data} \tag{1}$$

$$S = f^S(x) \qquad \triangleleft \quad \textbf{student's representation} \tag{2}$$

$$T = f^T(x) \qquad \triangleleft \quad \textbf{teacher's representation} \tag{3}$$

Intuitively speaking, we will consider the joint distribution $p(S, T)$ and the product of marginal distributions $p(S)p(T)$, so that, by maximizing KL divergence between these distributions, we can maximize the *mutual information* between student and teacher representations. To setup an appropriate loss that can achieve this aim, let us define a distribution $q$ with latent variable $C$ which decides whether a tuple $(f^T(x_i), f^S(x_j))$ was drawn from the joint ($C = 1$) or product of marginals ($C = 0$):

$$q(T, S|C = 1) = p(T, S), \quad q(T, S|C = 0) = p(T)p(S) \tag{4}$$

Now, suppose in our data, we are given 1 congruent pair (drawn from the joint distribution, i.e. the same input provided to $T$ and $S$) for every $N$ incongruent pairs (drawn from the product of marginals; independent randomly drawn inputs provided to $T$ and $S$). Then the priors on the latent $C$ are:

$$q(C = 1) = \frac{1}{N + 1}, \quad q(C = 0) = \frac{N}{N + 1} \tag{5}$$

By simple manipulation and Bayes' rule, the posterior for class $C = 1$ is given by:

$$q(C = 1|T, S) = \frac{q(T, S|C = 1)q(C = 1)}{q(T, S|C = 0)q(C = 0) + q(x, y|C = 1)q(C = 1)} \tag{6}$$

$$= \frac{p(T, S)}{p(T, S) + Np(T)p(S)} \tag{7}$$

Next, we observe a connection to mutual information as follows:

$$\log q(C = 1|T, S) = \log \frac{p(T, S)}{p(T, S) + Np(T)p(S)}$$

$$= -\log(1 + N\frac{p(T)p(S)}{p(T, S)}) \leq -\log(N) + \log \frac{p(T, S)}{p(T)p(S)} \qquad (8)$$

Then taking expectation on both sides w.r.t. $p(T, S)$ (equivalently w.r.t. $q(T, S|C = 1)$) and rearranging, gives us:

$$I(T; S) \geq \log(N) + \mathbb{E}_{q(T,S|C=1)} \log q(C = 1|T, S) \qquad \triangleleft \quad \textbf{MI bound} \qquad (9)$$

where $I(T; S)$ is the mutual information between the distributions of the teacher and student embeddings. Thus maximizing $\mathbb{E}_{q(T,S|C=1)} \log q(C = 1|T, S)$ w.r.t. the parameters of the student network $S$ increases a lower bound on mutual information. However, we do not know the true distribution $q(C = 1|T, S)$; instead we estimate it by fitting a model $h : \{\mathcal{T}, \mathcal{S}\} \to [0, 1]$ to samples from the data distribution $q(C = 1|T, S)$, where $\mathcal{T}$ and $\mathcal{S}$ represent the domains of the embeddings. We maximize the log likelihood of the data under this model (a binary classification problem):

$$\mathcal{L}_{critic}(h) = \mathbb{E}_{q(T,S|C=1)}[\log h(T, S)] + N\mathbb{E}_{q(T,S|C=0)}[1 - \log(h(T, S))] \qquad (10)$$

$$h^* = \arg\max_h \mathcal{L}_{critic}(h) \qquad \triangleleft \quad \textbf{optimal critic} \qquad (11)$$

We term $h$ the *critic* since we will be learning representations that optimize the critic's score. Assuming sufficiently expressive $h$, $h^*(T, S) = q(C = 1|T, S)$ (via Gibbs' inequality; see Sec. 6.2.1 for proof), so we can rewrite Eq. 9 in terms of $h^*$:

$$I(T; S) \geq \log(N) + \mathbb{E}_{q(T,S|C=1)}[\log h^*(T, S)] \qquad (12)$$

Therefore, we see that the optimal critic is an estimator whose expectation lower-bounds mutual information. We wish to learn a student that maximizes the mutual information between its representation and the teacher's, suggesting the following optimization problem:

$$f^{S*} = \arg\max_{f^S} \mathbb{E}_{q(T,S|C=1)}[\log h^*(T, S)] \qquad (13)$$

An apparent difficulty here is that the optimal critic $h^*$ depends on the current student. We can circumvent this difficulty by weakening the bound in (12) to:

$$I(T; S) \geq \log(N) + \mathbb{E}_{q(T,S|C=1)}[\log h^*(T, S)] + N\mathbb{E}_{q(T,S|C=0)}[\log(1 - h^*(T, S))] \qquad (14)$$

$$= \log(N) + \mathcal{L}_{critic}(h^*) = \log(N) + \max_h \mathcal{L}_{critic}(h) \qquad (15)$$

$$\geq \log(N) + \mathcal{L}_{critic}(h) \qquad (16)$$

The first line comes about by simply adding $N\mathbb{E}_{q(T,S|C=0)}[\log(1 - h^*(T, S))]$ to the bound in (12). This term is strictly negative, so the inequality holds. The last line follows from the fact that $\mathcal{L}_{critic}(h^*)$ upper-bounds $\mathcal{L}_{critic}(h)$. Optimizing (15) w.r.t. the student we have:

$$f^{S*} = \arg\max_{f^S} \max_h \mathcal{L}_{critic}(h) \qquad \triangleleft \quad \textbf{our final learning problem} \qquad (17)$$

$$= \arg\max_{f^S} \max_h \mathbb{E}_{q(T,S|C=1)}[\log h(T, S)] + N\mathbb{E}_{q(T,S|C=0)}[\log(1 - h(T, S))] \qquad (18)$$

which demonstrates that we may jointly optimize $f^S$ at the same time as we learn $h$. We note that due to (16), $f^{S*} = \arg\max_{f^S} \mathcal{L}_{critic}(h)$, for any $h$, *also* is a representation that optimizes a lower-bound (a weaker one) on mutual information, so our formulation does not rely on $h$ being optimized perfectly.

We may choose to represent $h$ with any family of functions that satisfy $h : \{\mathcal{T}, \mathcal{S}\} \to [0, 1]$. In practice, we use the following:

$$h(T, S) = \frac{e^{g^T(T)'g^S(S)/\tau}}{e^{g^T(T)'g^S(S)/\tau} + \frac{N}{M}} \qquad (19)$$

where $M$ is the cardinality of the dataset and $\tau$ is a temperature that adjusts the concentration level. In practice, since the dimensionality of $S$ and $T$ may be different, $g^S$ and $g^T$ linearly transform them

into the same dimension and further normalize them by $\mathcal{L}$-2 norm before the inner product. The form of Eq. (18) is inspired by NCE (Gutmann & Hyvärinen, 2010; Wu et al., 2018). Our formulation is similar to the InfoNCE loss (Oord et al., 2018) in that we maximize a lower bound on the mutual information. However we use a different objective and bound, which in our experiments we found to be more effective than InfoNCE.

**Implementation.** Theoretically, larger $N$ in Eq. 16 leads to tighter lower bound on MI. In practice, to avoid using very large batch size, we follow Wu et al. (2018) and implement a memory buffer that stores latent features of each data sample computed from previous batches. Therefore, during training we can efficiently retrieve a large number of negative samples from the memory buffer.

### 3.2 KNOWLEDGE DISTILLATION OBJECTIVE

The knowledge distillation loss was proposed in Hinton et al. (2015). In addition to the regular cross-entropy loss between the student output $y^S$ and one-hot label $y$, it asks the student network output to be as similar as possible to the teacher output by minimizing the cross-entropy between their output probabilities. The complete objective is:

$$\mathcal{L}_{KD} = (1 - \alpha)H(y, y^S) + \alpha\rho^2 H(\sigma(z^T/\rho), \sigma(z^S/\rho)) \tag{20}$$

where $\rho$ is the temperature, $\alpha$ is a balancing weight, and $\sigma$ is softmax function. $H(\sigma(z^T/\rho), \sigma(z^S/\rho))$ is further decomposed into $KL(\sigma(z^T/\rho)|\sigma(z^S/\rho))$ and a constant entropy $H(\sigma(z^T/\rho))$.

### 3.3 CROSS-MODAL TRANSFER LOSS

In the cross-modal transfer task shown in Fig. 1(b), a teacher network is trained on a source modality $\mathcal{X}$ with large-scale labeled dataset. We then wish to transfer the knowledge to a student network, but adapt it to another dataset or modality $\mathcal{Y}$. But the features of the teacher network are still valuable to help with learning of the student on another domain. In this transfer task, we use the contrastive loss Eq. 10 to match the features of the student and teacher. Additionally, we also consider other distillation objectives, such as KD discussed in previous section, Attention Transfer Zagoruyko & Komodakis (2016a) and FitNet Romero et al. (2014). Such transfer is conducted on a paired but unlabeled dataset $D = \{(x_i, y_i)|i = 1, ..., L, x_i \in \mathcal{X}, y_i \in \mathcal{Y}\}$. In this scenario, there is no true label $y$ of such data for the original training task on the source modality, and therefore we ignore the $H(y, y^S)$ term in all objectives that we test. Prior cross-modal work Aytar et al. (2016); Hoffman et al. (2016b;a) uses either $L_2$ regression or KL-divergence.

### 3.4 ENSEMBLE DISTILLATION LOSS

In the case of ensemble distillation shown in 1(c), we have $M > 1$ teacher networks, $f^{T_i}$ and one student network $f^S$. We adopt the contrastive framework by defining multiple pair-wise contrastive losses between features of each teacher network $f^{T_i}$ and the student network $f^S$. These losses are summed together to give the final loss (to be minimized):

$$\mathcal{L}_{CRD-EN} = H(y, y^S) - \beta \sum_i \mathcal{L}_{critic}(T_i, S) \tag{21}$$

## 4 EXPERIMENTS

We evaluate our contrastive representation distillation (CRD) framework in three knowledge distillation tasks: (a) model compression of a large network to a smaller one; (b) cross-modal knowledge transfer; (c) ensemble distillation from a group of teachers to a single student network.

**Datasets** (1) *CIFAR-100* (Krizhevsky & Hinton, 2009) contains 50K training images with 0.5K images per class and 10K test images. (2) *ImageNet* (Deng et al., 2009) provides 1.2 million images from 1K classes for training and 50K for validation. (3) *STL-10* (Coates et al., 2011) consists of a training set of 5K labeled images from 10 classes and 100K unlabeled images, and a test set of 8K images. (4) *TinyImageNet* (Deng et al., 2009) has 200 classes, each with 500 training images and 50 validaton images. (5) *NYU-Depth V2* (Silberman et al., 2012) consists of 1449 indoor images, each labeled with dense depth image and semantic map.

| Teacher
Student | WRN-40-2
WRN-16-2 | WRN-40-2
WRN-40-1 | resnet56
resnet20 | resnet110
resnet20 | resnet110
resnet32 | resnet32x4
resnet8x4 | vgg13
vgg8 |
|---|---|---|---|---|---|---|---|
| Teacher | 75.61 | 75.61 | 72.34 | 74.31 | 74.31 | 79.42 | 74.64 |
| Student | 73.26 | 71.98 | 69.06 | 69.06 | 71.14 | 72.50 | 70.36 |
| KD* | 74.92 | 73.54 | 70.66 | 70.67 | 73.08 | 73.33 | 72.98 |
| FitNet* | 73.58 (↓) | 72.24 (↓) | 69.21 (↓) | 68.99 (↓) | 71.06 (↓) | 73.50 (↑) | 71.02 (↓) |
| AT | 74.08 (↓) | 72.77 (↓) | 70.55 (↓) | 70.22 (↓) | 72.31 (↓) | 73.44 (↑) | 71.43 (↓) |
| SP | 73.83 (↓) | 72.43 (↓) | 69.67 (↓) | 70.04 (↓) | 72.69 (↓) | 72.94 (↓) | 72.68 (↓) |
| CC | 73.56 (↓) | 72.21 (↓) | 69.63 (↓) | 69.48 (↓) | 71.48 (↓) | 72.97 (↓) | 70.71 (↓) |
| VID | 74.11 (↓) | 73.30 (↓) | 70.38 (↓) | 70.16 (↓) | 72.61 (↓) | 73.09 (↓) | 71.23 (↓) |
| RKD | 73.35 (↓) | 72.22 (↓) | 69.61 (↓) | 69.25 (↓) | 71.82 (↓) | 71.90 (↓) | 71.48 (↓) |
| PKT | 74.54 (↓) | 73.45 (↓) | 70.34 (↓) | 70.25 (↓) | 72.61 (↓) | 73.64 (↑) | 72.88 (↓) |
| AB | 72.50 (↓) | 72.38 (↓) | 69.47 (↓) | 69.53 (↓) | 70.98 (↓) | 73.17 (↓) | 70.94 (↓) |
| FT* | 73.25 (↓) | 71.59 (↓) | 69.84 (↓) | 70.22 (↓) | 72.37 (↓) | 72.86 (↓) | 70.58 (↓) |
| FSP* | 72.91 (↓) | n/a | 69.95 (↓) | 70.11 (↓) | 71.89 (↓) | 72.62 (↓) | 70.23 (↓) |
| NST* | 73.68 (↓) | 72.24 (↓) | 69.60 (↓) | 69.53 (↓) | 71.96 (↓) | 73.30 (↓) | 71.53 (↓) |
| CRD | **75.48** (↑) | **74.14** (↑) | **71.16** (↑) | **71.46** (↑) | **73.48** (↑) | **75.51** (↑) | **73.94** (↑) |
| CRD+KD | 75.64 (↑) | 74.38 (↑) | 71.63 (↑) | 71.56 (↑) | 73.75 (↑) | 75.46 (↑) | 74.29 (↑) |

Table 1: Test *accuracy* (%) of student networks on CIFAR100 of a number of distillation methods (ours is CRD); see Appendix for citations of other methods. ↑ denotes outperformance over KD and ↓ denotes underperformance. We note that CRD is the *only* method to always outperform KD (and also outperforms all other methods). We denote by * methods where we used our reimplementation based on the paper; for all other methods we used author-provided or author-verified code. Average over 5 runs.

| Teacher
Student | vgg13
MobileNetV2 | ResNet50
MobileNetV2 | ResNet50
vgg8 | resnet32x4
ShuffleNetV1 | resnet32x4
ShuffleNetV2 | WRN-40-2
ShuffleNetV1 |
|---|---|---|---|---|---|---|
| Teacher | 74.64 | 79.34 | 79.34 | 79.42 | 79.42 | 75.61 |
| Student | 64.6 | 64.6 | 70.36 | 70.5 | 71.82 | 70.5 |
| KD* | 67.37 | 67.35 | 73.81 | 74.07 | 74.45 | 74.83 |
| FitNet* | 64.14 (↓) | 63.16 (↓) | 70.69 (↓) | 73.59 (↓) | 73.54 (↓) | 73.73 (↓) |
| AT | 59.40 (↓) | 58.58 (↓) | 71.84 (↓) | 71.73 (↓) | 72.73 (↓) | 73.32 (↓) |
| SP | 66.30 (↓) | 68.08 (↑) | 73.34 (↓) | 73.48 (↓) | 74.56 (↑) | 74.52 (↓) |
| CC | 64.86 (↓) | 65.43 (↓) | 70.25 (↓) | 71.14 (↓) | 71.29 (↓) | 71.38 (↓) |
| VID | 65.56 (↓) | 67.57 (↑) | 70.30 (↓) | 73.38 (↓) | 73.40 (↓) | 73.61 (↓) |
| RKD | 64.52 (↓) | 64.43 (↓) | 71.50 (↓) | 72.28 (↓) | 73.21 (↓) | 72.21 (↓) |
| PKT | 67.13 (↓) | 66.52 (↓) | 73.01 (↓) | 74.10 (↑) | 74.69 (↑) | 73.89 (↓) |
| AB | 66.06 (↓) | 67.20 (↓) | 70.65 (↓) | 73.55 (↓) | 74.31 (↓) | 73.34 (↓) |
| FT* | 61.78 (↓) | 60.99 (↓) | 70.29 (↓) | 71.75 (↓) | 72.50 (↓) | 72.03 (↓) |
| NST* | 58.16 (↓) | 64.96 (↓) | 71.28 (↓) | 74.12 (↑) | 74.68 (↑) | 74.89 (↑) |
| CRD | **69.73** (↑) | **69.11** (↑) | **74.30** (↑) | **75.11** (↑) | **75.65** (↑) | **76.05** (↑) |
| CRD+KD | 69.94 (↑) | 69.54 (↑) | 74.58 (↑) | 75.12 (↑) | 76.05 (↑) | 76.27 (↑) |

Table 2: Top-1 test *accuracy* (%) of student networks on CIFAR100 of a number of distillation methods (ours is CRD) for transfer across very different teacher and student architectures. CRD outperforms KD and all other methods. Importantly, some methods that require very similar student and teacher architectures perform quite poorly. E.g. FSP (Yim et al., 2017) cannot even be applied; AT (Ba & Caruana, 2014) and FitNet (Zagoruyko & Komodakis, 2016a) perform very poorly etc. We denote by * methods where we used our reimplementation based on the paper; for all other methods we used author-provided or author-verified code. Average over 3 runs.

## 4.1 Model Compression

**Setup** We experiment on CIFAR-100 and ImageNet with student-teacher combinations of various capacity, such as ResNet (He et al., 2016) or Wide ResNet (WRN) (Zagoruyko & Komodakis, 2016b).

**Results on CIFAR100** Table 1 and Table 2 compare top-1 *accuracies* of different distillation objectives (for details, see Section 6.1). Table 1 investigates students and teachers of the same architectural style, while Table 2 focuses on students and teachers from different architectures. We observe that our

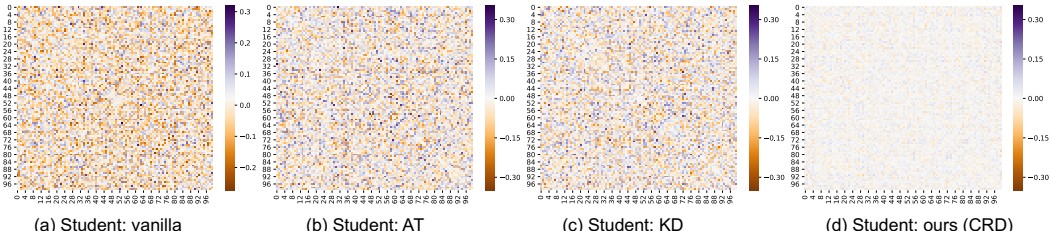

|   | (a) Student: vanilla | (b) Student: AT | (c) Student: KD | (d) Student: ours (CRD) |

Figure 2: The correlations between class logits of a teacher network are ignored by regular cross-entropy. Distillation frameworks use "soft targets" (Hinton et al., 2015) which effectively capture such correlations and transfer them to the student network, leading to the success of distillation. We visualize here the *difference* of correlation matrices of student and teacher logits, for different student networks on a CIFAR-100 knowledge distillation task: (a) Student trained without distillation, showing that the teacher and student cross-correlations are very different; (b) Student distilled by attention transfer (Zagoruyko & Komodakis, 2016a); showing reduced difference (see axis); (c) Student distilled by KL divergence (Hinton et al., 2015), also showing reduced difference; (d) Student distilled by our contrastive objective, showing significant matching between student's and teacher's correlations. In this visualization, we use WRN-40-2 as teacher and WRN-40-1 as student.

|       | Teacher | Student | AT    | KD    | SP    | CC    | Online KD * | CRD   | CRD+KD    |
|-------|---------|---------|-------|-------|-------|-------|-------------|-------|-----------|
| Top-1 | 26.69   | 30.25   | 29.30 | 29.34 | 29.38 | 30.04 | 29.45       | 28.83 | **28.62** |
| Top-5 | 8.58    | 10.93   | 10.00 | 10.12 | 10.20 | 10.83 | 10.41       | 9.87  | **9.51**  |

Table 3: Top-1 and Top-5 error rates (%) of student network ResNet-18 on ImageNet validation set. We use ResNet-34 released by PyTorch team as our teacher network, and follow the standard training practice of ImageNet on PyTorch except that we train for 10 more epochs. We compare our CRD with KD (Hinton et al., 2015), AT (Zagoruyko & Komodakis, 2016a) and Online-KD (Lan et al., 2018). "*" reported by the original paper Lan et al. (2018) using an ensemble of online ResNets as teacher, no pretrained ResNet-34 was used.

loss, which we call CRD (Contrastive Representation Distillation), consistently outperforms all other distillation objectives, including the original KD (an average relative improvement of 57%). Surprisingly, we found that KD works pretty well and none of the other methods consistently outperforms KD on their own. Another observation is that, while switching the teacher student combinations from same to different architectural styles, methods that distill intermediate representations tend to perform worse than methods that distill from the last several layers. For example, the Attention Transfer (AT) and FitNet methods even underperform the vanilla student. In contrast, PKT, SP and CRD that operate on last several layers performs well. This might because, architectures of different styles have their own solution paths mapping from the input to the output, and enforcing the mimic of intermediate representations thus might conflict with such inductive bias.

**Capturing inter-class correlations.** In Fig. 2, we compute the difference between the correlation matrices of the teacher's and student's logits; for three different students: vanilla student without distillation, trained by AT, KD or CRD (our method). It is clear that the CRD objective captures the most correlation structure in the logit as shown by the smaller differences between teacher and student. This is reflected in reduced error rates.

**Results on ImageNet** For a fair comparison with Zagoruyko & Komodakis (2016a) and Lan et al. (2018), we adopt the models from these papers, ResNet-34 as the teacher and ResNet-18 as the student. As shown in Table 3, the gap of top-1 accuracy between the teacher and student is 3.56%. The AT method reduces this gap by 0.95%, while ours narrow it by 1.42%, a 50% relative improvement. Results on ImageNet validates the scalability of our CRD.

**Transferability of representations** We are interested in *representations*, and a primary goal of representation learning is to acquire *general* knowledge, that is, knowledge that transfers to tasks or datasets that were unseen during training. Therefore, we test if the representations we distill transfer well. A WRN-16-2 student either distills from a WRN-40-2 teacher, or is trained from scratch on CIFAR100. The student serves as a frozen representation extractor (the layer prior to the logit) for images from STL-10 or TinyImageNet (all images downsampled to 32x32). We then train a *linear* classifier to perform 10-way (for STL-10) or 200-way (for TinyImageNet) classification to quantify the transferability of the representations. We compare CRD with multiple baselines such as KD and

|  | Student | KD | AT | FitNet | CRD | CRD+KD | Teacher |
|---|---|---|---|---|---|---|---|
| CIFAR100→STL-10 | 69.7 | 70.9 | 70.7 | 70.3 | 71.6 | **72.2** | 68.6 |
| CIFAR100→TinyImageNet | 33.7 | 33.9 | 34.2 | 33.5 | **35.6** | 35.5 | 31.5 |

Table 4: We transfer the representation learned from CIFAR100 to STL-10 and TinyImageNet datasets by freezing the network and training a linear classifier on top of the last feature layer to perform 10-way (STL-10) or 200-way (TinyImageNet) classification. For this experiment, we use the combination of teacher network WRN-40-2 and student network WRN-16-2. Classification accuracies (%) are reported.

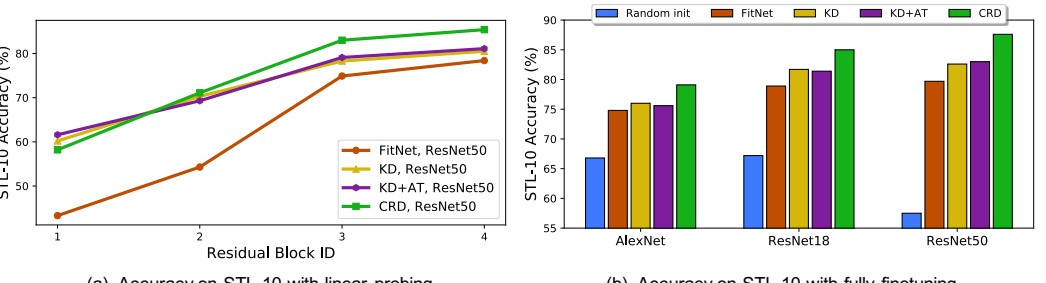

(a) Accuracy on STL-10 with linear probing

(b) Accuracy on STL-10 with fully finetuning

Figure 3: Top-1 classification accuracy on STL-10 using *chrominance* image ($ab$ channel in $Lab$ color space). We initialize the *chrominance* network randomly or by distilling from a *luminance* network, trained with large-scale labeled images. We evaluate distillation performance by (a) linear probing and (b) fully finetuning.

AT in Table 4. In this setting, all distillation methods except FitNet improve transferability of the learned representations on both STL-10 and TinyImageNet. While the teacher performs the best on the original CIFAR100 dataset, its representations transfer the worst to the other two datasets. This is perhaps the teacher's representations are biased towards the original task. Surprisingly, the student with CRD+KD distillation not only matches its teacher on CIFAR100 (see Table 4), but also transfers much better than the teacher, e.g., 3.6% improvement (STL-10) and 4.1% on TinyImageNet.

## 4.2 Cross-modal transfer

We consider a practical setting where modality $\mathcal{X}$ has large amount of labeled data while modality $\mathcal{Y}$ does not. Transfering knowledge from $\mathcal{X}$ to $\mathcal{Y}$ is a common challenge. For example, while large-scale RGB datasets are easily accessible, other modalities such as depth images are much harder to label at scale but have wide applications. We demonstrate the potential of CRD for cross-modal transfer in two scenarios: (a) transfer from luminance to chrominance; (b) transfer from RGB to depth images.

**Transferring from Luminance to Chrominace.** We work on $Lab$ color space, where $L$ represents Luminance and $ab$ Chrominance. We first train an $L$ network on TinyImageNet with supervision. Then we transfer knowledge from this $L$ network to $ab$ network on the unlabeled set of STL-10 with different objectives, including FitNet, KD, KD+AT and CRD. For convenience, we use the same architecture for student and teacher (they can also be different). Finally, we evaluate the knowlege of $ab$ network by two means: (1) *linear probing*: we freeze the $ab$ network and train a linear classifier on top of features from different layers to perform 10-way classification on STL-10 $ab$ images. This is a common practice (Alain & Bengio, 2016; Zhang et al., 2017) to evaluate the quality of network representations; (b) *fully finetuning*: we fully finetune the $ab$ network to obtain the best accuracy. We also use as a baseline the $ab$ network that is randomly initialized rather than distilled. Architectures investigated include AlexNet, ResNet-18 and ResNet-50. The results shown in Figure 3 show that CRD is more efficient for transferring inter-modal knowledge than other methods. Besides, we also note KD+AT does not improve upon KD, possibly because attention of luminance and chrominance are different and harder to transfer.

**Transferring from RGB to Depth.** We transfer the knowledge of a ResNet-18 teacher pretrained on ImageNet to a 5-layer student CNN operating on depth images. We follow a similar transferring procedure on NYU-Depth training set, except that we use a trick of contrasting between local and global features, proposed by Hjelm et al. (2018), to overcome the problem of insufficient data samples in the depth domain. Then the student network is further trained to predict semantic segmentation

| Metric (%) | Random Init. | KD | KD+AT | FitNet | CRD |
|---|---|---|---|---|---|
| Pix. Acc. | 56.4 | 58.9 | 60.1 | 60.8 | **61.6** |
| mIoU | 35.8 | 38.0 | 39.5 | 40.7 | **41.8** |

Table 5: Performance on the task of using depth to predict semantic segmentation labels. We initialize the depth network either randomly or by distilling from a ImageNet pre-pretrained ResNet-18 teacher.

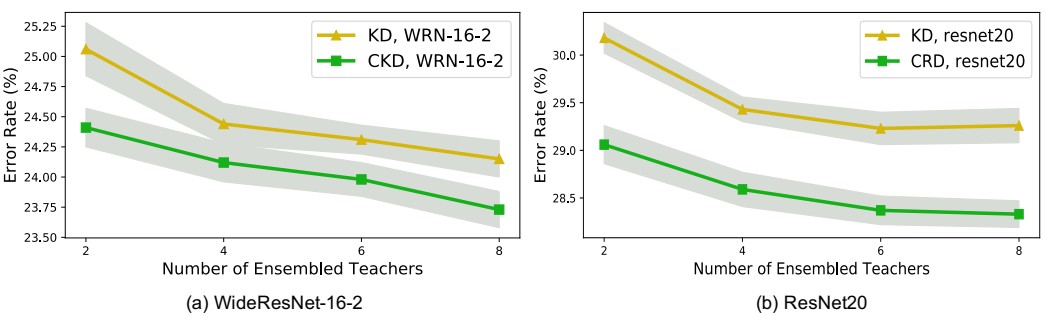

(a) WideResNet-16-2

(b) ResNet20

Figure 4: Distillation from an ensemble of teachers. We vary the number of ensembled teachers and compare KD with our CRD by using (a) WRN-16-2 and (b) ResNet20. Our CRD consistently achieves lower error rate.

maps from depth images. We note that both knowledge transfer and downstream training are conducted on the same set of images, i.e., the training set. Table 5 reports the average pixel prediction accuracy and mean Intersection-over-Union across all classes. All distillation methods can transfer knowledge from the RGB ResNet to the depth CNN. FitNet surpasses KD and KD+AT. CRD significantly outperforms all other methods.

## 4.3 DISTILLATION FROM AN ENSEMBLE

Better classification performance is often achieved by ensembles of deep networks, but these are usually too expensive for inference time, and distillation into a single network is a desirable task. We investigate the KL-divergence based KD and our CRD for this task, using the loss of Sec. 21. The network structures of each teacher and student are identical here, but an ensemble of multiple teachers can still provide rich knowledge for the student. To compare between KD and CRD on CIFAR100 dataset, we use WRN-16-2 and ResNet-20, whose single model error rates are 26.7% and 30.9% respectively. The results of distillation are presented in Figure 4, where we vary the number of ensembled teachers. CRD with 8 teachers decreases the error rate of WRN-16-2 to 23.7% and ResNet20 to 28.3%. In addition, CRD works consistently better than KD in all settings we test. These observations suggest that CRD is capable of distilling an ensemble of models into a single one which performs significantly better than a model of the same size that is trained from scratch.

## 4.4 ABLATIVE STUDY

**InfoNCE v.s. Ours.** InfoNCE (Oord et al., 2018) is an alternative contrastive objective which select a single positive out from a set of distractors via a softmax function. We compare InfoNCE with our contrastive objective Eq 18, when using the same number of negatives. The last two rows in Table 6 show that our objective outperforms InfoNCE in 4 out of 5 teacher-student combinations.

**Negative Sampling.** In this paper, we consider two negative sampling plocies when giving an anchor $x_i$: (1) $x_{j_{j \neq i}}$ for the unsupervised case when we have no labels, or (2) $x_{j_{y_j \neq y_i}}$ for supervised case, where $y_i$ represents the label associated with sample $x_i$. One might imagine that the first sampling strategy will increase the intra-class variance as we may push apart positives and negatives from the same underlying class, while the second would not. We quantitatively measure and report the gap between these two strategies in Table 6, which shows that the classification accuracy by the second sampling strategy is 0.81% higher for our objective and 0.62% higher for InfoNCE than the first one.

| sampling | objective | WRN-40-2 WRN-16-2 | resnet110 resnet20 | resnet110 resnet32 | resnet32x4 resnet8x4 | vgg13 vgg8 |
|---|---|---|---|---|---|---|
| $i \neq j$ | InfoNCE | 74.78 | 70.56 | 72.67 | 74.69 | 73.24 |
| | Ours | 74.48 | 70.64 | 72.64 | 74.67 | 73.39 |
| $y_i \neq y_j$ | InfoNCE | 75.15 | 71.39 | **73.53** | 75.22 | 73.74 |
| | Ours | **75.48** | **71.46** | 73.48 | **75.51** | **73.94** |

Table 6:  Ablative study of different contrastive objectives and negative sampling policies on CIFAR100. For contrastive objectives, we compare our objective with InfoNCE (Oord et al., 2018); For negative sampling policy, when given an anchor image $x_i$ from the dataset, we consider either randomly sample negative $x_j$ such that (a) $i \neq j$, or (b) $y_i \neq y_j$ where $y$ represents the class label. Average over 5 runs.

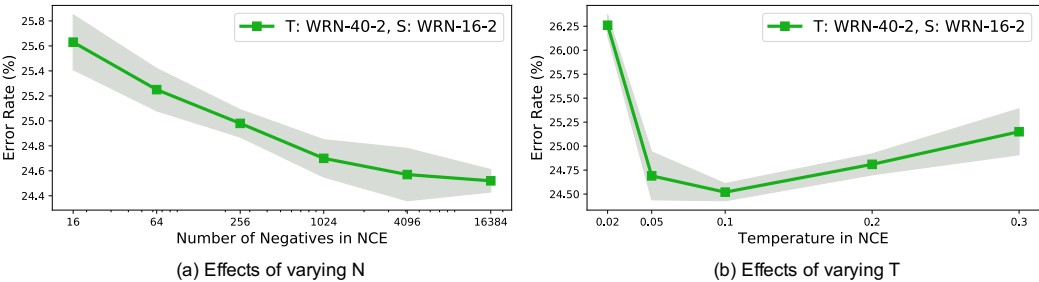

(a) Effects of varying N          (b) Effects of varying T

Figure 5:  Effects of varying the number of negatives, shown in (a), or the temperature, shown in (b).

### 4.5 Hyper-parameters and computation overhead

There are two main hyper-parameters in our contrastive objectives: (1) the number of negative samples $N$ in Eq. 18, and (2) temperature $\tau$ which modulates the softmax probability. We adopt WRN-40-2 as teacher and WRN-16-2 as student for parameter analysis. Experiments are conducted on CIFAR100 and the results are shown in Figure 5.

**Number of negatives** $N$ We have validated different $N$: $16, 64, 256, 1024, 4096, 16384$. As shown in Figure 5(a), increasing $N$ leads to improved performance. However, the difference of error rate between $N = 4096$ and $N = 16384$ is less than $0.1\%$. Therefore, we use $N = 16384$ for reporting the accuracy while in practice $N = 4096$ should suffice.

**Temperature** $\tau$ We varied $\tau$ between 0.02 and 0.3. As Figure 5(b) illustrates, both extremely high or low temperature lead to a sub-optimal solution. In general, temperatures between 0.05 and 0.2 work well on CIFAR100. All experiments but those on ImageNet use a temperature of 0.1. For ImageNet, we use $\tau = 0.07$. The optimal temperature may vary across different datasets and require further tuning.

**Computational Cost** We use ResNet-18 on ImageNet for illustration. CRD uses extra 260 MFLOPs, which is about 12% of the original 2 GFLOPs. In practice, we did not notice significant difference of training time on ImageNet (e.g., 1.75 epochs/hour v.s. 1.67 epochs/hour on two Titan-V GPUs). The memory bank for storing all 128-d features of ImageNet only costs around 600MB memory, and therefore we store it on GPU memory.

## 5 Conclusion

We have developed a novel technique for neural network distillation, using the concept of contrastive objectives, which are usually used for representation learning. We experimented with our objective on a number of applications such as model compression, cross-modal transfer and ensemble distillation, outperforming other distillation objectives by significant margins in all these tasks. Our contrastive objective is the only distillation objective that consistently outperforms knowledge distillation across a wide variety of knowledge transfer tasks. Prior objectives only surpass KD *when combined* with KD. Contrastive learning is a simple and effective objective with practical benefits.

**Acknowledgments.**   We thank Baoyun Peng for providing the code of CC (Peng et al., 2019) and Frederick Tung for verifying our reimplementation of SP (Tung & Mori, 2019). This research was supported in part by Google Cloud and iFlytek.

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

# 6 APPENDIX

## 6.1 OTHER METHODS

We compare to the following other state-of-the-art methods from the literature:

1. Knowledge Distillation (KD) (Hinton et al., 2015)
2. Fitnets: Hints for thin deep nets (Romero et al., 2014)
3. Attention Transfer (AT) (Zagoruyko & Komodakis, 2016a)
4. Similarity-Preserving Knowledge Distillation (SP) (Tung & Mori, 2019);
5. Correlation Congruence (CC) (Peng et al., 2019)
6. Variational information distillation for knowledge transfer (VID) (Ahn et al., 2019)
7. Relational Knowledge Distillation (RKD) (Park et al., 2019)
8. Learning deep representations with probabilistic knowledge transfer (PKT) (Passalis & Tefas, 2018)
9. Knowledge transfer via distillation of activation boundaries formed by hidden neurons (AB) (Heo et al., 2019)
10. Paraphrasing complex network: Network compression via factor transfer (FT) (Kim et al., 2018)
11. A gift from knowledge distillation: Fast optimization, network minimization and transfer learning (FSP) (Yim et al., 2017)
12. Like what you like: Knowledge distill via neuron selectivity transfer (NST) (Huang & Wang, 2017)

## 6.2 CONTRASTIVE LOSS – DETAILS

### 6.2.1 PROOF THAT $h^*(T, S) = q(C = 1|T, S)$

We wish to model some true distribution $q(C|T = t, S = s)$. $C$ is a binary variable, so we can model $q(C|T = t, S = s)$ as a Bernoulli distribution with a single parameter $h(S = s, T = t) \in [0, 1]$,

defining for convenience $h'(C = 1, S = s, T = t) = h(S = s, T = t)$ and $h'(C = 0, S = s, T = t) = 1 - h(S = s, T = t)$. The log likelihood function is:

$$\mathbb{E}_{c \sim q(C|S=s,T=t)}[\log h'(C = c, S = s, T = t)] \tag{22}$$

By Gibbs' inequality, the max likelihood fit is $h'(C = c, S = s, T = t) = q(C = c|S = s, T = t)$, which also implies that $h(S = s, T = t) = q(C = 1|S = s, T = t)$.

We now demonstrate that our objective in Eq. (10) is proportional to a summation over terms Eq. (22) for all $s \in \mathcal{S}$ and $t \in \mathcal{T}$.

$$\mathbb{E}_{s,t \sim q(S,T)}[\mathbb{E}_{c \sim q(C|S=s,T=t)}[\log h'(C = c, S = s, T = t)]] \tag{23}$$

$$= \mathbb{E}_{c,s,t \sim q(C,S,T)}[\log h'(C = c, S = s, T = t)]] \tag{24}$$

$$= \mathbb{E}_{s,t \sim q(S,T|C=1)q(C=1)}[\log h(S = s, T = t)] +$$
$$\mathbb{E}_{s,t \sim q(S,T|C=0)q(C=0)}[\log(1 - h(S = s, T = t))] \tag{25}$$

$$= \frac{1}{N + 1}\mathbb{E}_{s,t \sim q(S,T|C=1)}[\log h(S = s, T = t)] +$$
$$\frac{N}{N + 1}\mathbb{E}_{s,t \sim q(S,T|C=0)}[\log(1 - h(S = s, T = t))] \tag{26}$$

Notice that (26) is proportional to Eq. (10) from the main paper. For sufficiently expressive $h$, then, each term inside the expectation in Eq. (23) can be maximized, resulting in $h^*(T = t, S = s) = q(C = 1|T = t, S = t)$ for all $s$ and $t$. $\square$

### 6.3 NETWORK ARCHITECTURES

Wide Residual Network (WRN) (Zagoruyko & Komodakis, 2016b). WRN-d-w represnets wide resnet with depth $d$ and width factor $w$.

resnet (He et al., 2016). We use resnet-d to represent **cifar**-style resnet with 3 groups of basic blocks, each with 16, 32, and 64 channels respectively. In our experiments, resnet8 x4 and resnet32 x4 indicate a 4 times wider network (namely, with 64, 128, and 256 channels for each of the block)

ResNet (He et al., 2016). ResNet-d represents **ImageNet**-style ResNet with Bottleneck blocks and more channels.

MobileNetV2 Sandler et al. (2018). In our experiments, we use a width multiplier of 0.5.

vgg (Simonyan & Zisserman, 2014). the vgg net used in our experiments are adapted from its original ImageNet counterpart.

ShuffleNetV1 (Zhang et al., 2018a), ShuffleNetV2 (Tan et al., 2019). ShuffleNets are proposed for efficient training and we adapt them to input of size 32x32.

### 6.4 IMPLEMENTATION DETAILS

All methods evaluated in our experiments use SGD.

For CIFAR-100, we initialize the learning rate as 0.05, and decay it by 0.1 every 30 epochs after the first 150 epochs until the last 240 epoch. For MobileNetV2, ShuffleNetV1 and ShuffleNetV2, we use a learning rate of 0.01 as this learning rate is optimal for these models in a grid search, while 0.05 is optimal for other models.

For ImageNet, we follow the standard PyTorch practice but train for 10 more epochs. Batch size is 64 for CIFAR-100 or 256 for ImageNet.

The student is trained by a combination of cross-entropy classification objective and a knowledge distillation objective, shown as follows:

$$\mathcal{L} = \mathcal{L}_{cross-entropy} + \beta \mathcal{L}_{distill} \tag{27}$$

For the weight balance factor $\beta$, we directly use the optimal value from the original paper if it is specified, or do a grid search with teacher WRN-40-2 and student WRN-16-2. This results in the following list of $\beta$ used for different objectives:

1. Fitnets (Romero et al., 2014): $\beta = 100$

2. AT (Zagoruyko & Komodakis, 2016a): $\beta = 1000$

3. SP (Tung & Mori, 2019): $\beta = 3000$

4. CC (Peng et al., 2019): $\beta = 0.02$

5. VID (Ahn et al., 2019): $\beta = 1$

6. RKD (Park et al., 2019): $\beta_1 = 25$ for distance and $\beta_2 = 50$ for angle. For this loss, we combine both term following the original paper.

7. PKT (Passalis & Tefas, 2018): $\beta = 30000$

8. AB (Heo et al., 2019): $\beta = 0$, distillation happens in a separate pre-training stage where only distillation objective applies.

9. FT (Kim et al., 2018): $\beta = 500$

10. FSP (Yim et al., 2017): $\beta = 0$, distillation happens in a separate pre-training stage where only distillation objective applies.

11. NST (Huang & Wang, 2017): $\beta = 50$

12. CRD: $\beta = 0.8$, in general $\beta \in [0.5, 1.5]$ works reasonably well.

For KD(Hinton et al., 2015), we follow Eq. 20 and set $\alpha = 0.9$ and $T = 4$.

### 6.5 VISUALIZATION OF THE CORRELATION DISCREPANCY

We visualize the correlation discrepancy for different distillation objectives across various combinations of student and teacher networks. As shown in Fig. 6, our contrastive distillation objective siginificantly outperforms other objectives, in terms of minimizing the correlation discrepancy between student and teacher networks. The normalized correlation coefficients are computed at the logit layer.

### 6.6 COMBINING DIFFERENT DISTILLATION OBJECTIVES

Table 7 shows the effect of combining various objectives with KD. Most methods is able to slightly outperform KD after this combination. We also the check the compatibility of our distillation objective with KD and PKT. The combination of CRD and KD/PKT further improves over single CRD objective.

### 6.7 ADDITIONAL RESULTS ON TRANSFERABILITY OF REPRESENTATIONS

In addtion to Table 4 shown in the main text, we also evaluate the transferabilit of the representations of various other distillation methods and summarize the results in Table 8. In general, CRD achieves the best transferring accuracy in 7 out of 10 settings.

### 6.8 DEEP MUTUAL LEARNING SETTING

In (Zhang et al., 2018b), a Deep Mutual Learning setting was proposed where the teacher and student networks are trained simultaneously rather than sequentially. The benefit is that not only the student networks but also the teacher models will be improved. Here we investigate the possibility of incorporating different distillation objectives into this training framework. The results are summarizied in Table 9. We observe that in general logit based distillation methods is better than non-logit based distillation methods in this setting, i.e., KD performs the best. We conjecture that during the mutual training phase, KD performs like an advanced label smoothing regularization and does not require the logit to be very accurate. But the feature-based distillation methods are hard to learn knowledge from the teacher model when it is handicapped. On the other hand, we notice that the combination of KD and CRD leads to better performance on the student side, as shown in the last row of Table 9.

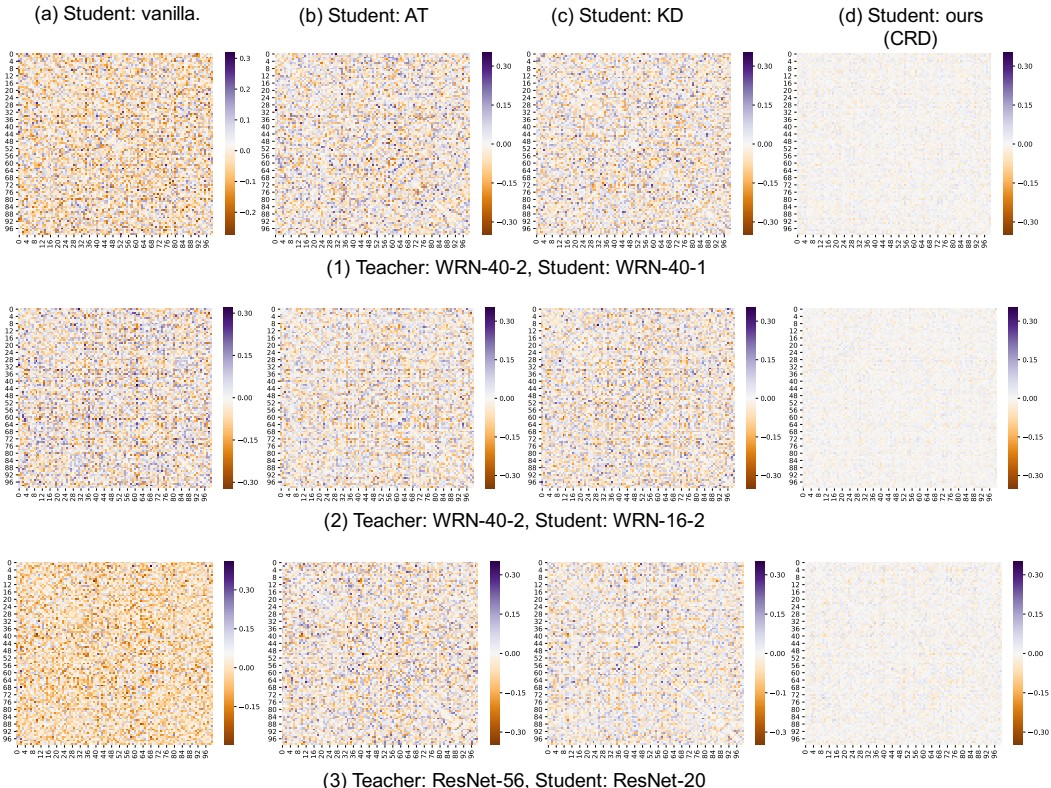

Figure 6: The correlations between class logits output by the teacher network show the "dark knowledge" Hinton et al. (2015) that must be transferred to a student networks. A student network that captures these correlations tends to perform better at the task. We visualize here the difference between correlation matrices of the student and teacher at the logits, for different student networks on a Cifar100 knowledge distillation task: (a) A student trained without distillation; (b) A student distilled by attention transfer Zagoruyko & Komodakis (2016a) (c) A student distilled by KL divergence Hinton et al. (2015); (d) A student distilled by our contrastive objective. Our objective greatly improves the structured knowledge (correlations) in the output units.

## 6.9    STANDARD DEVIATION FOR RESULTS ON CIFAR-100 BENCHMARK

The standard deviation over multiple runs on CIFAR-100 benchmark is provided in Table 10 for student and teacher models of the same architectural type, and in Table 11 for student and teacher models of different architectural types.

| Teacher
Student | WRN-40-2
WRN-16-2 | resnet110
resnet20 | resnet32x4
resnet8x4 | vgg13
vgg8 | vgg13
MobileNetV2 | ResNet50
vgg8 | resnet32x4
ShuffleNetV2 |
|---|---|---|---|---|---|---|---|
| Teacher | 75.61 | 74.31 | 79.42 | 74.64 | 74.64 | 79.34 | 79.42 |
| Student | 73.26 | 69.06 | 72.50 | 70.36 | 64.60 | 70.36 | 71.82 |
| KD | 74.92 | 70.67 | 73.33 | 72.98 | 67.37 | 73.81 | 74.45 |
| FitNet+KD | 75.12 (↑) | 70.67 (↑) | 74.66 (↑) | 73.22 (↑) | 66.90 (↓) | 73.24 (↓) | 75.15 (↑) |
| AT+KD | 75.32 (↑) | 70.97 (↑) | 74.53 (↑) | 73.48 (↑) | 65.13 (↓) | 74.01 (↑) | 75.39 (↑) |
| SP+KD | 74.98 (↑) | 71.02 (↑) | 74.02 (↑) | 73.49 (↑) | 68.41 (↑) | 73.52 (↓) | 74.88 (↑) |
| CC+KD | 75.09 (↑) | 70.88 (↑) | 74.21 (↑) | 73.04 (↑) | 68.02 (↑) | 73.48 (↓) | 74.71 (↑) |
| VID+KD | 75.14 (↑) | 71.10 (↑) | 74.56 (↑) | 73.19 (↑) | 68.27 (↑) | 73.46 (↓) | 74.85 (↑) |
| RKD+KD | 74.89 (↓) | 70.77 (↑) | 73.79 (↑) | 72.97 (↓) | 67.87 (↑) | 73.51 (↓) | 74.55 (↑) |
| PKT+KD | 75.33 (↑) | 70.72 (↑) | 74.23 (↑) | 73.25 (↑) | 68.13 (↑) | 73.61 (↓) | 74.66 (↑) |
| AB+KD | 70.27 (↓) | 70.97 (↑) | 74.40 (↑) | 73.35 (↑) | 68.23 (↑) | 73.65 (↓) | 74.99 (↑) |
| FT+KD | 75.15 (↑) | 70.88 (↑) | 74.62 (↑) | 73.44 (↑) | 66.99 (↓) | 72.98 (↓) | 75.06 (↑) |
| NST+KD | 74.67 (↓) | 71.01 (↑) | 74.28 (↑) | 73.33 (↑) | 63.77 (↓) | 71.74 (↓) | 75.24 (↑) |
| CRD | 75.48 (↑) | 71.46 (↑) | 75.51 (↑) | 73.94 (↑) | 69.73 (↑) | 74.30 (↑) | 75.65 (↑) |
| CRD+KD | 75.64 (↑) | 71.56 (↑) | 75.46 (↑) | 74.29 (↑) | **69.94** (↑) | 74.58 (↑) | **76.05** (↑) |
| CRD+PKT | **75.91** (↑) | **71.65** (↑) | **75.90** (↑) | **74.57** (↑) | 69.59 (↑) | **74.68** (↑) | 75.94 (↑) |

Table 7: Test *accuracy* (%) of student networks on CIFAR100 of combining distillation methods with KD; we check the compatibility of our objective with KD as well as PKT. ↑ denotes outperformance over KD and ↓ denotes underperformance.

| Teacher
Student | WRN-40-2
WRN-16-2 | | resnet56
resnet20 | | resnet32x4
resnet8x4 | | vgg13
MobileNetV2 | | resnet32x4
ShuffleNetV2 | |
|---|---|---|---|---|---|---|---|---|---|---|
| dataset | STL | TI | STL | TI | STL | TI | STL | TI | STL | TI |
| Vanilla | 69.7 | 33.7 | 69.1 | 30.4 | 71.8 | 36.7 | 64.2 | 29.1 | 65.1 | 28.9 |
| KD | 70.9 | 33.9 | 69.1 | 32.0 | 71.8 | 36.2 | 66.3 | 29.9 | 69.5 | 33.4 |
| FitNet | 70.3 | 33.5 | 68.1 | 29.5 | 73.5 | 40.2 | 66.9 | 31.4 | 71.7 | 36.2 |
| AT | 70.7 | 34.2 | 70.1 | 31.0 | 74.1 | 39.2 | 66.3 | 28.4 | 72.5 | 37.3 |
| SP | 71.3 | 34.1 | 68.0 | 30.2 | 72.9 | 37.6 | 67.8 | 31.6 | 68.8 | 33.4 |
| CC | 70.5 | 33.7 | 68.6 | 30.4 | 72.1 | 37.1 | 66.3 | 30.6 | 69.9 | 34.4 |
| VID | 71.0 | 34.6 | 69.9 | 32.4 | 73.9 | 39.6 | 66.1 | 31.1 | 71.4 | 36.5 |
| RKD | 71.6 | 35.1 | 70.6 | 32.6 | 73.9 | 38.8 | 68.4 | 32.9 | 71.5 | 37.8 |
| PKT | 71.6 | 34.8 | 69.1 | 31.6 | 73.5 | 38.0 | 69.5 | 33.5 | 70.7 | 36.4 |
| AB | 70.8 | 32.2 | 68.5 | 29.7 | 73.4 | 37.6 | 67.6 | 31.0 | 71.3 | 36.2 |
| FT | 71.7 | 34.7 | **70.9** | 33.5 | **75.2** | **40.4** | 68.9 | 32.5 | 72.9 | 38.7 |
| FSP | 69.6 | 33.6 | 68.1 | 30.4 | 72.4 | 37.1 | n/a | n/a | n/a | n/a |
| NST | 70.2 | 32.6 | 68.5 | 30.1 | 73.9 | 38.7 | 64.9 | 27.8 | 72.3 | 37.8 |
| CRD | 71.6 | **35.6** | 70.2 | **34.3** | 74.8 | 40.2 | **71.6** | **35.7** | **73.5** | **40.1** |
| CRD+KD | **72.2** | 35.5 | 70.5 | 34.1 | 74.6 | 39.3 | 70.7 | 34.6 | 73.1 | 39.9 |

Table 8: We measure the transferability of the student network, by evaluating a linear classifier on top of its frozen representations on STL10 (abbreviated as "STL") and TinyImageNet (abbreviated as "TI"). The best accuracy is bolded and the second best is underlined.

| Teacher
Student | WRN-40-2
WRN-16-2 | WRN-40-2
WRN-40-1 | resnet56
resnet20 | resnet110
resnet20 | resnet110
resnet32 | resnet32x4
resnet8x4 | vgg13
vgg8 | ResNet50
vgg8 | |
|---|---|---|---|---|---|---|---|---|---|
| Vanilla | 75.61 | 75.61 | 72.34 | 74.31 | 74.31 | 79.42 | 74.64 | 79.34 | |
| KD | 77.89 | **77.42** | **74.51** | **75.95** | **76.70** | **79.90** | **76.53** | **79.92** | |
| AT | 76.02 | 76.42 | 72.44 | 73.72 | 73.69 | 79.08 | 73.21 | 79.33 | |
| SP | 76.36 | 76.08 | 73.11 | 73.58 | 73.81 | 79.50 | 75.75 | 73.34 | T |
| CC | 76.42 | 76.30 | 71.95 | 72.78 | 74.19 | 79.44 | 74.80 | 77.76 | |
| CRD | 77.30 | 77.23 | 73.25 | 75.32 | 74.37 | 79.81 | 75.25 | 78.65 | |
| CRD+KD | **78.01** | 77.39 | 73.86 | 75.31 | 75.53 | 79.36 | 77.23 | 79.04 | |
| Vanilla | 73.26 | 71.98 | 69.06 | 69.06 | 71.14 | 72.50 | 70.36 | 70.36 | |
| KD | 74.98 | 73.66 | 70.85 | 70.71 | **73.24** | 74.77 | 73.39 | 74.00 | |
| AT | 73.04 | 71.57 | 69.20 | 69.04 | 70.87 | 72.32 | 69.55 | 69.56 | |
| SP | 72.82 | 71.47 | 69.45 | 69.31 | 70.75 | 72.63 | 70.57 | 70.66 | S |
| CC | 73.19 | 71.46 | 68.82 | 69.51 | 71.24 | 72.57 | 71.12 | 70.36 | |
| CRD | 75.22 | 73.53 | 70.92 | 70.80 | 72.65 | 75.24 | 73.19 | 73.21 | |
| CRD+KD | **75.89** | **74.12** | **70.90** | **71.60** | 73.07 | 75.34 | **74.08** | **74.22** | |

Table 9: Test *accuracy* (%) of student and teacher networks on CIFAR100 with the Deep Mutual Training Zhang et al. (2018b) setting, where the teacher and student networks are trained simultaneously rather than sequentially. We use "T" and "S" to denote the teacher and student models, respectively.

| Teacher
Student | WRN-40-2
WRN-16-2 | WRN-40-2
WRN-40-1 | resnet56
resnet20 | resnet110
resnet20 | resnet110
resnet32 | resnet32x4
resnet8x4 | vgg13
vgg8 |
|---|---|---|---|---|---|---|---|
| Teacher
Student | 75.61
73.26 | 75.61
71.98 | 72.34
69.06 | 74.31
69.06 | 74.31
71.14 | 79.42
72.50 | 74.64
70.36 |
| KD | 74.92 ± 0.28 | 73.54 ± 0.20 | 70.66 ± 0.24 | 70.67 ± 0.27 | 73.08 ± 0.18 | 73.33 ± 0.25 | 72.98 ± 0.19 |
| FitNet | 73.58 ± 0.32 | 72.24 ± 0.24 | 69.21 ± 0.36 | 68.99 ± 0.27 | 71.06 ± 0.13 | 73.50 ± 0.28 | 71.02 ± 0.31 |
| AT | 74.08 ± 0.25 | 72.77 ± 0.10 | 70.55 ± 0.27 | 70.22 ± 0.16 | 72.31 ± 0.08 | 73.44 ± 0.19 | 71.43 ± 0.09 |
| SP | 73.83 ± 0.12 | 72.43 ± 0.27 | 69.67 ± 0.20 | 70.04 ± 0.21 | 72.69 ± 0.41 | 72.94 ± 0.23 | 72.68 ± 0.19 |
| CC | 73.56 ± 0.26 | 72.21 ± 0.25 | 69.63 ± 0.32 | 69.48 ± 0.19 | 71.48 ± 0.21 | 72.97 ± 0.17 | 70.71 ± 0.24 |
| VID | 74.11 ± 0.24 | 73.30 ± 0.13 | 70.38 ± 0.14 | 70.16 ± 0.39 | 72.61 ± 0.28 | 73.09 ± 0.21 | 71.23 ± 0.06 |
| RKD | 73.35 ± 0.09 | 72.22 ± 0.20 | 69.61 ± 0.06 | 69.25 ± 0.05 | 71.82 ± 0.34 | 71.90 ± 0.11 | 71.48 ± 0.05 |
| PKT | 74.54 ± 0.04 | 73.45 ± 0.19 | 70.34 ± 0.04 | 70.25 ± 0.04 | 72.61 ± 0.17 | 73.64 ± 0.18 | 72.88 ± 0.09 |
| AB | 72.50 ± 0.26 | 72.38 ± 0.31 | 69.47 ± 0.09 | 69.53 ± 0.16 | 70.98 ± 0.39 | 73.17 ± 0.31 | 70.94 ± 0.18 |
| FT | 73.25 ± 0.20 | 71.59 ± 0.15 | 69.84 ± 0.12 | 70.22 ± 0.10 | 72.37 ± 0.31 | 72.86 ± 0.12 | 70.58 ± 0.08 |
| FSP | 72.91 ± 0.24 | n/a | 69.95 ± 0.21 | 70.11 ± 0.05 | 71.89 ± 0.11 | 72.62 ± 0.13 | 70.23 ± 0.23 |
| NST | 73.68 ± 0.11 | 72.24 ± 0.22 | 69.60 ± 0.13 | 69.53 ± 0.15 | 71.96 ± 0.07 | 73.30 ± 0.28 | 71.53 ± 0.13 |
| CRD | 75.48 ± 0.09 | 74.14 ± 0.22 | 71.16 ± 0.17 | 71.46 ± 0.09 | 73.48 ± 0.13 | 75.51 ± 0.18 | 73.94 ± 0.22 |
| CRD+KD | 75.64 ± 0.21 | 74.38 ± 0.11 | 71.63 ± 0.15 | 71.56 ± 0.16 | 73.75 ± 0.24 | 75.46 ± 0.25 | 74.29 ± 0.12 |

Table 10: Test *accuracy* (%) of student networks on CIFAR100 of a number of distillation methods (ours is CRD). Standard deviation is provided.

| Teacher
Student | vgg13
MobileNetV2 | ResNet50
MobileNetV2 | ResNet50
vgg8 | resnet32x4
ShuffleNetV1 | resnet32x4
ShuffleNetV2 | WRN-40-2
ShuffleNetV1 |
|---|---|---|---|---|---|---|
| Teacher | 74.64 | 79.34 | 79.34 | 79.42 | 79.42 | 75.61 |
| Student | 64.6 | 64.6 | 70.36 | 70.5 | 71.82 | 70.5 |
| KD | $67.37 \pm 0.32$ | $67.35 \pm 0.32$ | $73.81 \pm 0.13$ | $74.07 \pm 0.19$ | $74.45 \pm 0.27$ | $74.83 \pm 0.17$ |
| FitNet | $64.14 \pm 0.50$ | $63.16 \pm 0.47$ | $70.69 \pm 0.22$ | $73.59 \pm 0.15$ | $73.54 \pm 0.22$ | $73.73 \pm 0.32$ |
| AT | $59.40 \pm 0.20$ | $58.58 \pm 0.54$ | $71.84 \pm 0.28$ | $71.73 \pm 0.31$ | $72.73 \pm 0.09$ | $73.32 \pm 0.35$ |
| SP | $66.30 \pm 0.38$ | $68.08 \pm 0.38$ | $73.34 \pm 0.34$ | $73.48 \pm 0.42$ | $74.56 \pm 0.22$ | $74.52 \pm 0.24$ |
| CC | $64.86 \pm 0.25$ | $65.43 \pm 0.15$ | $70.25 \pm 0.12$ | $71.14 \pm 0.06$ | $71.29 \pm 0.38$ | $71.38 \pm 0.25$ |
| VID | $65.56 \pm 0.42$ | $67.57 \pm 0.28$ | $70.30 \pm 0.31$ | $73.38 \pm 0.09$ | $73.40 \pm 0.17$ | $73.61 \pm 0.12$ |
| RKD | $64.52 \pm 0.45$ | $64.43 \pm 0.42$ | $71.50 \pm 0.07$ | $72.28 \pm 0.39$ | $73.21 \pm 0.28$ | $72.21 \pm 0.16$ |
| PKT | $67.13 \pm 0.30$ | $66.52 \pm 0.33$ | $73.01 \pm 0.14$ | $74.10 \pm 0.25$ | $74.69 \pm 0.34$ | $73.89 \pm 0.16$ |
| AB | $66.06 \pm 0.48$ | $67.20 \pm 0.37$ | $70.65 \pm 0.09$ | $73.55 \pm 0.31$ | $74.31 \pm 0.11$ | $73.34 \pm 0.09$ |
| FT | $61.78 \pm 0.33$ | $60.99 \pm 0.37$ | $70.29 \pm 0.19$ | $71.75 \pm 0.20$ | $72.50 \pm 0.15$ | $72.03 \pm 0.16$ |
| NST | $58.16 \pm 0.26$ | $64.96 \pm 0.44$ | $71.28 \pm 0.13$ | $74.12 \pm 0.19$ | $74.68 \pm 0.26$ | $74.89 \pm 0.25$ |
| CRD | $69.73 \pm 0.42$ | $69.11 \pm 0.28$ | $74.30 \pm 0.14$ | $75.11 \pm 0.32$ | $75.65 \pm 0.10$ | $76.05 \pm 0.14$ |
| CRD+KD | $69.94 \pm 0.05$ | $69.54 \pm 0.39$ | $74.58 \pm 0.27$ | $75.12 \pm 0.35$ | $76.05 \pm 0.09$ | $76.27 \pm 0.29$ |

Table 11: Top-1 test *accuracy* (%) of student networks on CIFAR100 of a number of distillation methods (ours is CRD) for transfer across very different teacher and student architectures. Standard deviation is provided.

