# OpenReview forum: "Contrastive Representation Distillation"
_ICLR.cc/2020/Conference — Accept (Poster)_

### Official Review · AnonReviewer3 · 2019-10-22
**Official Blind Review #3**

**Rating:** 3

**Review:**

I do not necessarily see something wrong with the paper, but I'm not convinced of the significance (or sufficient novelty) of the approach.

The way I understand it, an independent assumption on internal representation is relaxed by capturing correlations between them and higher order dependencies between them using a different objective function(distance measure), the problem then becomes nothing but trying to minimize another distance metric between teacher and student networks on an intermediate layer.

Comparing with original distillation method (KD) I'm not sure how significant the improvement is. And technically this is just a distance metric between the internal representation or output of the student and teacher. Sure it is a more involved distance metric, however it is in the spirit of what the distillation work is all about and I do not see this as being fundamentally different, or at least not different enough for an ICLR paper.

The experimental results also suggest only a marginal improvement compared to other methods. It would be helpful to also include the variance of each experiment i.e., it was mentioned that the results were averaged by repeating 5 experiments to make sure the proposed method consistently better than others.  Rightnow, It is hard to compare with other approaches. The paper gives off a feeling that this method is not novel. Why this particular distance metric between the representations? Why not just L2?


**Experience Assessment:**

I do not know much about this area.

**Review Assessment: Checking Correctness Of Derivations And Theory:**

I assessed the sensibility of the derivations and theory.

**Review Assessment: Checking Correctness Of Experiments:**

I carefully checked the experiments.

**Review Assessment: Thoroughness In Paper Reading:**

I made a quick assessment of this paper.

---

> ### Author Response · Authors · 2019-11-14
> **Response to Reviewer 3**
>
> Dear Reviewer 3,
>
> Thank you for your feedback.
>
> “the problem then becomes nothing but trying to minimize another distance metric between teacher and student networks on an intermediate layer”
> Different distance metrics can make a big difference. Theoretically, we show our objective is maximizing a lower bound on mutual information between the latent representations of the teacher and student network. Empirically, we show our method consistently outperforms all 12 recent knowledge distillation methods published in NeurIPS, CVPR, ICCV and other top conferences (we note that many of these previous works were also just different distance metrics between teacher and student networks).
>
> “Comparing with original distillation method (KD) I'm not sure how significant the improvement is”
> We clarify that it’s important to see the relative improvement, rather than the absolute improvement. Suppose the gap of accuracy between well-trained teacher and student networks is only 3 percentage points, how can we ask the knowledge from the teacher model improves the student by 10 absolute percentage points, or even 5 percentage points. Indeed, our CRD outperforms the second best method (KD, see Table 1&2) with a 57% average relative improvement.
>
> “however it is in the spirit of what the distillation work is all about and I do not see this as being fundamentally different, or at least not different enough for an ICLR paper”
> These 12 other methods we compare here are just as similar to KD but are published in top venues. However, our method not only outperforms recent SOTA methods, but also forges a connection between two literatures that have evolved mostly independently: knowledge distillation and representation learning.
>
> “It would be helpful to also include the variance of each experiment”
> We have added the standard deviation in Sec 6.9 and Tables 9&10.
>
> “Why this particular distance metric between the representations? Why not just L2?”
> We have both theoretical analysis and empirical validation for the improvement provided by this distance metric. An L2 metric is provided by FitNet, which we compare to in Tables 1, 2, 4, 5 and Figure 3 (CRD outperforms FitNet).

---

### Official Review · AnonReviewer1 · 2019-10-23
**Official Blind Review #1**

**Rating:** 6

**Review:**

This paper combines a contrastive objective measuring the mutual information between the representations learned by a teacher and a student networks for model distillation. The objective enforces correlations between the learned representations. When combined with the popular KL divergence between the predictions of the two networks, the proposed model shows consistently improvement over existing alternatives on three distillation tasks.

This is a solid work – it is based on sound principles and provides both rigorous theoretical analysis and extensive empirical evidence. I only have two minor suggestions.

1, From Section 3.2 to Section 3.4, it is not clear to me that on the model compression task, are both the proposed contrastive loss and the loss in Eq. (10) used?

2, The “Deep mutual learning”, Zhang et al, CVPR’18 paper needs to be discussed. I’d also like to see some experiments on the effects of training the teacher and student networks jointly from scratch using the proposed loss.


**Experience Assessment:**

I have published one or two papers in this area.

**Review Assessment: Checking Correctness Of Derivations And Theory:**

I assessed the sensibility of the derivations and theory.

**Review Assessment: Checking Correctness Of Experiments:**

I assessed the sensibility of the experiments.

**Review Assessment: Thoroughness In Paper Reading:**

I made a quick assessment of this paper.

---

> ### Author Response · Authors · 2019-11-14
> **Response to Reviewer 1**
>
>
> Dear Reviewer 1,
>
> Thank you very much for the constructive comments.
>
> 1. For our CRD results in Table 1&2, we only use CRD loss. We have added CRD+KD results in the revised version to avoid such confusion, and CRD+KD shows further small improvement.
>
> 2. We have added a discussion of the “Deep Mutual Learning” paper in Sec 6.8. We have evaluated several different distillation methods under the mutual learning setting, see Table 8.
>
> Please don’t hesitate to let us know for any further feedback. Thanks!

---

### Official Review · AnonReviewer2 · 2019-10-28
**Official Blind Review #2**

**Rating:** 6

**Review:**

Summary & Pros
- This paper proposes a well-principled distillation method based on contrastive loss maximizing the mutual information between teacher and student models.
- This paper provides extensive experiments that demonstrate the effectiveness of the proposed method. The performance gap compared to the existing distillation approaches seem to be significant.

Major Concerns:
- The authors claim that "none of the other methods consistently outperform KD on their own". I feel that this claim is somewhat aggressive because some of them outperform KD without combining with KD, e.g., Table 1 in FT (Kim et al., 2018) and Table 2 in SP (Tung & Mori, 2019). Since the distillation (or transfer) methods are typically sensitive to hyperparameters (e.g., architecture types, transfer connections between layers), I also wonder how to set the hyperparameters for baselines, especially in Table 2, because choosing the transfer connections between different architectures is very important when using feature-based methods such as FitNet and AT.
- Moreover, the baselines are developed for improving distillation performance, not replacing KD. Especially, feature-based methods (e.g., FitNet, NST) are easily combined with logit-based ones (e.g., KD). So I think the compatibility between the proposed and exisiting methods should be checked. However, in this paper, only Table 4 shows the compatibility with KD (CRD+KD).
- The authors compare the proposed method with only KD, AT, FitNet except Table 1-3. For example, when evaluating the transferability (Table 4), why other baselines are not compared?
- VID also maximizes MI between penultimate layers. What is the key difference and why CRD perform better? I think detailed verfication should be provided in the paper.

Minor Comments
- Comparison with Online-KD is unfair because it does not use pre-trained ResNet32.
- Why only use penultimate layers? CRD between intermediate layers is also available like VID.
- A result in Table 1 is missing (FSP WRN40-2 -> WRN40-1).
- In Section 4, both CKD (contrastive knowledge distillation) and CRD (contrastive representation distillation) are used, so one of them should be removed.
- In Section 4.1 Transferability paragraph, "Table 3" should be changed to "Table 4".
- On page 4, "space" after "before the inner product." should be removed.

I think the proposed method is well-principled and provides meaningful improvements on various distillation settings, thus this paper seems to be above the borderline. It would be better if additional supports for the above concerns is provided in a rebuttal.

**Experience Assessment:**

I have published one or two papers in this area.

**Review Assessment: Checking Correctness Of Derivations And Theory:**

I assessed the sensibility of the derivations and theory.

**Review Assessment: Checking Correctness Of Experiments:**

I assessed the sensibility of the experiments.

**Review Assessment: Thoroughness In Paper Reading:**

I read the paper at least twice and used my best judgement in assessing the paper.

---

> ### Author Response · Authors · 2019-11-14
> **Response to Reviewer 2**
>
>
> Dear Reviewer 2,
>
> Thank you for the very constructive comments.
>
> 1. We have revised the contribution section to make it more accurate.
>
> Hyperparameters (architecture types, transfer connections between layers). For architecture types or student-teach pairs, we just randomly sampled over a candidate pool which contains widely-used networks. As for the transfer connections for feature based distillation methods, we tried two methods: (1) connect the layers with 0.25, 0.5 and 0.75 depth of the whole network; (2) connect the last layer of each specific spatial size, e.g., the last layers, which have spatial size of 32, 16, 8 and 4, are connected correspondingly. We found (2) generally works better and stick to it.
>
> We agree that we might achieve higher accuracies if we separately tune hyperparameters for each specific student-teacher pairs. But on the other hand, we consider it to be more interesting to see how each method can generalize across different architecture types without much tuning. This is also our motivation of randomly sampling different student-teacher pairs to form the benchmark.
>
> 2. Compatibility. We have included another table in the revised paper (see Sec. 6.6 and Table 6) discussing the combination of different distillation objectives. We found: (1) other objectives combined with KD still underperform CRD; (2) combining CRD with other objectives, such as KD and PKT, can further improve the performance.
>
> 3. The main reason why we don’t extensively compare with all other baselines in other Tables is that we have limited resources. Given such a situation, we are inclined to distribute our resources towards building complete results (Tables 1&2) on the standard CIFAR-100 benchmark, rather than the proof of concept on specific settings, such as Table 5 and Figure 3.
>
> We appreciate the idea of comparing transferability with other methods. We have included a new table (see Sec 6.7 and Table 7) showing the transferability of all methods with different architectures.
>
> 4. VID also claims to capture the MI between the representations of student and teacher, but their instantiation is very different from ours. In order to achieve a tractable computation of MI, VID used a Gaussian distribution as a variational approximation of the true conditional distribution. Our conjecture is that the true conditional distribution is perhaps very different from the Gaussian assumption, and therefore it leads to suboptimal results.
> Other issues:
> We have added a note to indicate that Online-kd does not use pre-trained ResNet-34. We can also remove this item if needed.
> We tried CRD on middle layers but only see very marginal improvement, so we opt to keep our methods simpler. Indeed, only using penultimate layers makes CRD more robust to different architecture types.
> FSP only works for teachers and students with the size of features being equal, so it’s not available there. We will mark it as N/A.
> The typos have been fixed.
>
> Please don’t hesitate to let us know for any additional comments. Thank you!

---

### Public Comment · ~Xiangyu_He1 · 2021-10-10
**A misleading typo in Equation(10)?**

It seems that $N\mathbb{E}_{q(T,S|C=0)}[1-\log(h(T,S))]$ is actually $N\mathbb{E}_{q(T,S|C=0)}[\log(1-h(T,S))]$.

---

### Decision · Program_Chairs · 2019-12-19

**Decision:**

Accept (Poster)

**Comment:**

This paper presents a new distillation method with theoretical and empirical supports.

Given reviewers' comments and AC's reading, the novelty/significance and application-scope shown in the paper can be arguably limited. However, the authors extensively verified and compared the proposed methods and existing ones by showing significant improvements under comprehensive experiments. As the distillation method can enjoy a broader usage, I think the propose method in this paper can be influential in the future works.

Hence, I think this is a borderlines paper toward acceptance.